# LUMIRAG: A UNIFIED MULTIMODAL RAG LARGE MODEL BRIDGING TEXT AND IMAGE RETRIEVAL

## ABSTRACT

Retrieval-Augmented Generation (RAG) enhances large language models' ability to leverage external knowledge. However, existing models remain limited in their unified understanding and generation of text and multimodal retrieved content. We present **LumiRAG**, a suite of Qwen2.5 based models achieving strong RAG capabilities across modalities through systematic fine-tuning with high-quality data. Our approach comprises three key components: (1) **Human-synthetic hybrid dataset** with adaptive domain harvesting, dual-source generation, and multi-layer quality control, producing 520K samples across text RAG, multimodal tasks, and expert-annotated dialogues; (2) **Three-stage progressive instruction tuning** that unifies supervised fine-tuning, context-augmented instruction tuning, and reinforcement learning with Optimized-DAPO for stepwise performance alignment; (3) **Cross-modal reinforcement learning framework** employing reward shaping and stabilized training to jointly optimize retrieval accuracy and generation quality. Extensive evaluations on ChatRAG-Bench, long-form summarization benchmarks (CNN/DailyMail, XSum), MMRAG-Bench, and MMTAB demonstrate that LUMIRAG substantially outperforms open-source and proprietary baselines, establishing new **state-of-the-art** performance across diverse modalities and task types. Model weights, datasets, and evaluation code will be open-sourced to support reproducibility and future research.

## 1 INTRODUCTION

The field of retrieval-augmented generation (RAG) has experienced rapid evolution with the emergence of large language models capable of integrating external knowledge sources (Lewis et al., 2020; Gao et al., 2023). While significant progress has been made in text-based RAG, with open-source models such as ChatQA-1.0-70B achieving competitive performance against proprietary systems (Liu et al., 2024a), the integration of visual and textual modalities remains challenging due to representation misalignment and training complexity (Zhu et al., 2024; Chen et al., 2025). Compounding this, developing unified multimodal RAG systems capable of matching state-of-the-art proprietary models remains an unresolved obstacle. Existing approaches face three major challenges: (i) **fragmented model training**, where textual and visual modules are trained independently, resulting in misaligned representations; (ii) **cross-modal data scarcity**, with a lack of high-quality conversational datasets spanning text and vision; and (iii) **incomplete evaluation frameworks**, which fail to account for factual consistency and cross-modal conflicts. To address these limitations, we introduce LumiRAG, a unified framework that substantially outperforms both open-source and proprietary baselines (Figure 1). Specifically, we make the following contributions:

1. **Hybrid Dataset Construction**: We propose a tri-source data integration strategy combining human-curated (63K, 9.0-9.2/10), synthetic (457K, 7.8-8.1/10), and open-source data (360K), yielding 880K training samples with 520K high-quality ones covering text RAG, multimodal tasks, and expert conversations.

2. **Three-Stage Progressive Training**: We design a systematic pipeline that gradually builds RAG capabilities: Stage 1 establishes foundational instruction-following; Stage 2 strengthens context integration; Stage 3 achieves human preference alignment via our DS-DAPO reinforcement learning algorithm, significantly outperforming standard baselines.

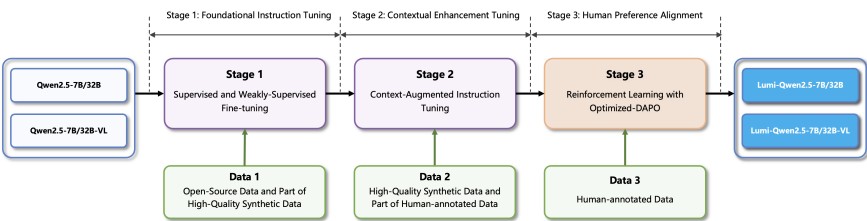

Figure 1: Training Pipeline

3. **Efficient Cross-Modal RL**: We introduce DS-DAPO (Dynamic Sampling through Resampling and Batch Completion for DAPO), which addresses gradient vanishing in uniformly correct outputs, achieving 52.91% training efficiency improvement over standard DAPO while jointly optimizing retrieval accuracy and generation quality.

4. **State-of-the-Art Results**: Our experiments demonstrate that **LumiRAG-Qwen2.5-7B (60.13)** outperforms the 70B parameter NV Llama3.1-ChatQA-1.5 (58.26) with 10× fewer parameters, while **LumiRAG-Qwen2.5-32B (64.06)** surpasses GPT-4o (50.54) and Claude3.5-Sonnet (43.05). On multimodal benchmarks, our approach achieves 66.68 average score with +25.2% accuracy gain and 28.7% hallucination reduction.

We discuss related work in § 2, covering conversational and multimodal RAG, supervised and instruction fine-tuning, and reinforcement learning optimization. We present our methodology in § 3, including training dataset, model supervised fine-tuning training, and model RL training. We conduct experiments in § 4, covering tasks and datasets, baseline models, benchmark results, and ablation studies. We present results in § 5, including main results (ChatRAG Bench, MMRAG, MRAG Bench), fine-grained analyses, and case studies. We conclude in § 6. All model weights, datasets, and evaluation code will be open-sourced to support reproducibility and advance unified multimodal RAG research.

## 2 RELATED WORK

We evaluate models on diverse benchmarks covering text, multimodal, and structured reasoning tasks. Large Language Models (LLMs) demonstrate strong capabilities but still face issues such as hallucination and limited factual grounding (Brown et al., 2020; Ji et al., 2023). Retrieval-Augmented Generation (RAG) addresses these limitations by incorporating external knowledge (Lewis et al., 2020). Here, we review key advances in conversational and multimodal RAG, as well as supervised fine-tuning and reinforcement learning optimization.

### 2.1 CONVERSATIONAL AND MULTIMODAL RAG

Recent advances in RAG emphasize context-aware retrieval, cross-modal integration, and reinforcement learning for multimodal consistency. Adaptive retrieval mechanisms, such as dynamic gating (Sheffield et al., 2024) and "think-before-retrieve" strategies (Guan et al., 2025), improve generation quality by selectively retrieving relevant information. Multimodal RAG combines visual and textual features in shared embedding spaces (Liu et al., 2024a) or structured graph-based representations (Zhang et al., 2023), enabling robust reasoning and comprehensive document understanding (Microsoft Research, 2025). Reinforcement learning methods like GRPO-CARE and DanceGRPO further enhance multimodal consistency, achieving significant gains in video QA and generation quality (Kim et al., 2024; Park et al., 2024).

### 2.2 EFFICIENT TRAINING AND OPTIMIZATION METHODS

Parameter-efficient methods like LoRA and selective layer freezing enable effective LLM adaptation at low cost (Hu et al., 2021). Instruction fine-tuning with RLHF allows smaller models to surpass larger ones (Ouyang et al., 2022), and dual-phase or mixed fine-tuning mitigates catastrophic forgetting across tasks (Wang et al., 2024). For reinforcement learning optimization, GRPO reduces

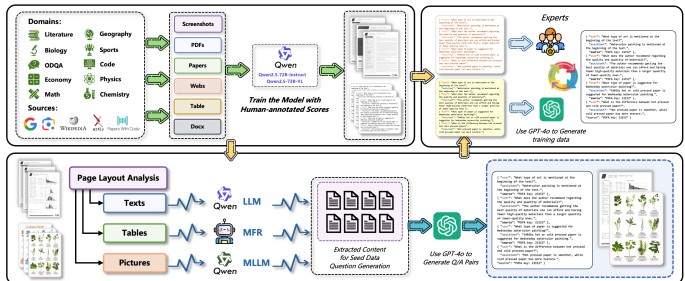

Figure 2: Shows the LumiRAG pipeline. Multi-domain, multi-format data are decomposed into text, tables, and images, processed by specialized modules for question generation. Progressive training with human and GPT-4o validation ensures quality and addresses fragmented training.

computational overhead via group-based comparisons (Shao et al., 2024), with extensions like Prefix Grouper and GiGPO improving scalability and long-horizon performance (Tang et al., 2024; Yao et al., 2024). DAPO achieves faster convergence and higher accuracy with dynamic sampling and clip-higher techniques (ByteDance, 2025; Raschka, 2025), while experience replay methods such as RePO and EIS-GRPO further enhance robustness (Chen et al., 2024; Li et al., 2024), forming the foundation for efficient, instruction-aligned multimodal RAG models.

## 3 METHOD

### 3.1 TRAINING DATASET

We construct a hybrid dataset integrating human annotations, synthetic generation, and open-source resources to support multimodal RAG training. It addresses the scarcity of high-quality supervision, modality imbalance, and the quality–scale trade-off, while ensuring both accuracy and broad coverage. The dataset (Table 1) comprises 880K samples, where human annotations provide reliable supervision, synthetic data adds scale and diversity, and open-source resources enhance robustness, collectively enabling strong multimodal reasoning and adaptability. Detailed description of the dataset see Appendix 6.

Table 1: Training Data Statistics

| Data Type | Category | Text-Only | Multimodal | Total | Quality Score |
|---|---|---|---|---|---|
| Human-annotated Data (SFT) | Fine-tuning | 25K | 20K | 45K | 9.2/10 |
| Synthetic Data (SFT) | Fine-tuning | 160K | 145K | 305K | 8.1/10 |
| Human-annotated Data (RL) | Preference | 10K | 8K | 18K | 9.0/10 |
| Synthetic Data (RL) | Preference | 70K | 82K | 152K | 7.8/10 |
| **Total** | - | 265K | 255K | **520K** | 8.0/10 |
| Open-source Data (SFT) | Fine-tuning | 200K | 160K | 360K | - |
| **Total** | - | 465K | 415K | **880K** | - |

### 3.1.1 SFT FINE-TUNING DATA

**Human-annotated Data**: We create **45K dialogues** (25K text-only, 20K multimodal) via expert role-playing. These dialogues feature multi-turn contextual progression, enabling gradual context accumulation. Multimodal samples are aligned with visual inputs (e.g., code snippets, diagrams, API screenshots), ensuring fine-grained grounding. The dataset has a quality score of 9.2/10 and serves as the foundation for supervised fine-tuning.

**Synthetic Data**: To address annotation bottlenecks, we employ a dual-stream generation framework across twelve technical domains as shown in Figure 2. A text stream (Qwen2.5-32B fine-tuned on curated data) and a multimodal stream (vision models for tables and diagrams) produce 305K samples (160K text-only, 145K multimodal, quality 8.1/10). The pipeline ensures semantic diversity, covers 95% of low-frequency concepts, and supports rapid domain expansion ($\sim$ 5K samples/hour).

**Open-source Data**. We also include 360K open-source dialogues (200K text-only, 160K multi-modal), enhancing domain coverage and adding diverse real-world data.

### 3.1.2 REINFORCEMENT LEARNING DATA

**Human-annotated Data**. We construct **18K preference pairs** (10K text-only, 8K multimodal; quality **9.0/10**), annotated by domain specialists. Evaluations capture nuanced distinctions in technical accuracy, reasoning depth, contextual coherence, and visual-textual alignment, offering high-quality supervision for RLHF.

**Synthetic Data**. We scale preference data via an **automated pipeline**(Figure 2) that generates **152K pairs** (70K text-only, 82K multimodal; quality **7.8/10**). Variants are created using temperature/nucleus sampling and constitutional prompting, then ranked by an ensemble of specialized evaluator models (for text) and vision-language evaluators (for multimodal). Principle-based filtering removes unsafe or biased responses. The pipeline sustains $\geq$ **8K preference pairs/hour**, ensuring scalability.

### 3.1.3 RETRIEVAL SYSTEM CONFIGURATION

Our training data employs source-specific retrieval strategies. **Open-source datasets** (360K samples) use pre-packaged question-context-answer triples from community benchmarks (e.g., ChatQA-Training-Data (Liu et al., 2024b), OK-VQA (Marino et al., 2019)). **Synthetic data** (457K samples) uses a BERT-base dual encoder (MS MARCO (Bajaj et al., 2018)) with FAISS indexing (Johnson et al., 2019) to retrieve Top-5 contexts from 5M documents across 12 domains. **Human-annotated data** (63K samples) is mixed: 40% retrieval-augmented, 60% direct conversations.

During **inference**, we follow benchmark protocols: ChatRAG Bench uses provided retrieval results, while comparative studies integrate into existing frameworks (Search-R1 (Jin et al., 2025) for text, EVA-CLIP-8B (Sun et al., 2023) for multimodal) with identical retrieval systems across all compared models. Human evaluation shows 78.3% Top-5 recall. Details in Appendix 6.

### 3.1.4 DATA QUALITY ASSESSMENT

We adopt a **multi-dimensional quality control framework** combining human-annotated data with high inter-annotator agreement (Cohen's $\kappa_C$ and Fleiss' $\kappa_F > 0.75$) and hierarchical expert review, synthetic data validated for factual, logical, and cross-modal consistency, Constitutional AI filtering for helpfulness, harmlessness, and honesty, and continuous feedback loops across annotation, training, and benchmarks to iteratively refine guidelines. A detailed Data Quality Assessment is provided in Appendix 6

### 3.2 SUPERVISED FINE-TUNING

We adopt a two-stage progressive fine-tuning strategy to develop RAG capabilities on Qwen2.5, inspired by instruction fine-tuning (Ouyang et al., 2022; Wang et al., 2022) and curriculum learning (Bengio et al., 2009; Hacohen & Weinshall, 2019).

**Training Configuration.** We train two types of models in parallel: text-only models (Qwen2.5-7B/32B) for pure text RAG and multimodal models (Qwen2.5-VL-7B/32B) for vision-language RAG. To accommodate both modalities within a unified training framework, we employ adaptive hyperparameters in our loss functions.

**Stage 1: Supervised & Weakly-Supervised Tuning.** Establishes instruction-following and multimodal understanding using 360K open-source and 150K synthetic samples. Training employs a learning rate of $2 \times 10^{-5}$, batch size 128, gradient accumulation 4, and curriculum learning from single-turn to multi-turn reasoning. Loss combines LM, cross-modal alignment, and contrastive modality consistency in equation 1:

$$\mathcal{L}_{\text{Stage1}} = \mathcal{L}_{\text{LM}} + \lambda_1 \mathcal{L}_{\text{align}} + \lambda_2 \mathcal{L}_{\text{modal}} \tag{1}$$

where $\mathcal{L}_{\text{LM}}$ denotes the language modeling loss, $\mathcal{L}_{\text{align}}$ represents cross-modal alignment loss, and $\mathcal{L}_{\text{modal}}$ captures modality consistency. Hyperparameter Settings. For text-only models (Qwen2.5),

$\lambda_1 = \lambda_2 = 0$ as no visual modality is involved; for multimodal models (Qwen2.5-VL), $\lambda_1 = 0.5$ and $\lambda_2 = 0.3$, tuned on validation sets to balance language understanding and cross-modal alignment.

Weak supervision uses self-training: for each instruction, N responses are sampled via nucleus sampling (temperature = 0.7, top_p = 0.9)(Holtzman et al., 2019), with the best sample selected via ensemble selection (Lee, 2013; Xie et al., 2020).

**Stage 2: Context-Enhanced Instruction Tuning.** Enhances RAG with 155K synthetic and 25K human-annotated samples, covering long-context, multi-hop reasoning, and cross-modal retrieval. A dynamic context gating function adjusts attention weights (Equation 2):

$$\alpha_i = \text{Gate}(\mathbf{q}, \mathbf{c_i}) = \sigma(W_g[\mathbf{q}; \mathbf{c_i}; \mathbf{q} \odot \mathbf{c_i}] + b_g) \qquad (2)$$

where $\mathbf{q} \in \mathbb{R}^d$ is the query representation (averaged user query encoding), $\mathbf{c}_i \in \mathbb{R}^d$ is the $i$-th document representation, $\mathbf{W}_g \in \mathbb{R}^{1 \times 3d}$ and $b_g \in \mathbb{R}$ are learnable parameters, and $\sigma(\cdot)$ is sigmoid activation. The gating coefficient $\alpha_i \in [0, 1]$ controls dependence on document $\mathbf{c}_i$ during generation, with $K = 5$ retrieved documents per query. These gating weights dynamically modulate cross-attention over retrieved documents, enabling focus on relevant contexts while suppressing noisy ones. The $\alpha_i$ values provide interpretability by indicating each document's importance.

The retrieval-aware objective combines LM, retrieval relevance, factual consistency (BERTScore) (Zhang et al., 2019), and ranking loss in Equation 3:

$$\mathcal{L}_{\text{Stage2}} = \mathcal{L}_{\text{LM}} + \lambda_3 \mathcal{L}_{\text{retrieval}} + \lambda_4 \mathcal{L}_{\text{consistency}} + \lambda_5 \mathcal{L}_{\text{relevance}} \qquad (3)$$

where $\mathcal{L}_{\text{retrieval}}$ aligns gating weights with semantic relevance scores computed by a pre-trained sentence encoder, encouraging the model to assign higher weights to more relevant documents; $\mathcal{L}_{\text{consistency}}$ enforces factual consistency via BERTScore (Zhang et al., 2020), measuring token-level semantic similarity between generated text and retrieved contexts to prevent hallucination; $\mathcal{L}_{\text{relevance}}$ applies pairwise ranking loss to optimize the relative ordering of gating weights based on document relevance signals. We set $\lambda_3 = 0.3$, $\lambda_4 = 0.4$, and $\lambda_5 = 0.2$ for both text-only and multimodal models. For multimodal models, the cross-modal alignment term from Stage 1 (with $\lambda_1 = 0.5$) is maintained throughout Stage 2 training to preserve vision-language consistency. Complete mathematical definitions of all loss terms are provided in Appendix 6.

**Optimization & Efficiency.** Stage 2 uses a reduced learning rate $1 \times 10^{-5}$, gradient clipping, FlashAttention-2 (Dao et al., 2022), gradient checkpointing (Chen et al., 2016), and ZeRO-2 sharding (Rajbhandari et al., 2020).

**Stability & Regularization.** Hierarchical importance sampling balances domains, modalities, and difficulty; convergence is monitored via validation perplexity, task F1, and human evaluation with early stopping. Label smoothing (Szegedy et al., 2016), weight decay (Loshchilov & Hutter, 2017), and attention dropout (Srivastava et al., 2014) enhance generalization.

This compact two-stage approach enables strong RAG capabilities while preserving core language understanding, achieving notable performance on both text and multimodal tasks with computational efficiency.

### 3.3 REINFORCEMENT LEARNING TRAINING

#### 3.3.1 OUR RL METHOD: DS-DAPO

In RL algorithms like GRPO (Shao et al., 2024), both fully correct and incorrect prompts yield zero advantage, causing vanishing gradients and stalled updates. Moreover, as high-accuracy samples accumulate during training, effective prompts per batch decrease, increasing gradient variance. While DAPO (Wang et al., 2024) employs *dynamic sampling* to address this issue, its oversampling strategy (e.g., $3\times$ batch size) incurs computational inefficiency. We propose **DS-DAPO** (*Dynamic Sampling through Resampling and Batch Completion for DAPO*), which retains DAPO's advantages while enhancing efficiency through three key improvements: exact batch sampling, uniform-accuracy prompt filtering, and pass rate computation: $P_j = (1/G) \sum_{i=1}^{G} \mathbb{I}(a_i = 1)$, where $G$ denotes the number of outputs sampled per prompt, and $\mathbb{I}(\cdot)$ is the indicator function. We sort prompts by descending pass

rate to prioritize those with higher correctness rates. Finally, prompts are resampled to complete the batch to the nearest multiple of the mini-batch size (e.g., expanding from 413 to 448 prompts when the mini-batch size is 64).

The reconstructed batch loss is then computed as equation 4 and equation 5:

$$\mathcal{J}^{\mathcal{B}_r}(\theta) = \mathbb{E}_{\mathcal{B}_r \sim \mathcal{D}, (q,a) \sim \mathcal{B}_r, \{o^i\}_{i=1}^G \sim \pi_{\theta_{old}}(\cdot|q)} \left[ \frac{1}{\sum_{i=1}^G |o^i|} \sum_{i=1}^G \sum_{t=1}^{|o^i|} L_{i,t}(\theta) \right], \tag{4}$$

$$L_{i,t}(\theta) = \min\left( r_{i,t}(\theta)\hat{A}_{i,t}, \; \text{clip}(r_{i,t}(\theta), 1 - \epsilon_{\text{low}}, 1 + \epsilon_{\text{high}})\hat{A}_{i,t} \right), \quad |\mathcal{B}_r| = k \cdot \mathcal{B}_{\text{mini}} \tag{5}$$

where $|\mathcal{B}_r|$ is the reconstructed batch size and $\mathcal{B}_{\text{mini}}$ is the mini-batch size. The detailed algorithmic procedure of DS-DAPO is presented in Algorithm 1.

---

**Algorithm 1** DS-DAPO

---

**Require:** Train batch $B$, Global batch size $gbs$ (e.g., 512), Mini batch size $mbs$ (e.g., 64)
**Ensure:** Train batch after resampling and completion $B_{resampled}$
1: Initialize prompts buffer $Q$ for filtered prompts
2: Initialize pass rates buffer $P_a$ for pass rates of filtered prompts ▷ Step 1: Filter train batch $B$
3: **for** each $q_j \in B$ **do**
4:     **if** all outputs of $q_j$ are correct or incorrect **then**
5:         filter $q_j$
6:     **else**
7:         $q_j \to Q$
8:         compute pass rate of $q_j$: $p_j = \frac{1}{G} \sum_{i=1}^G \mathbb{I}(a_i = 1)$
9:         $p_j \to P_a$
10:     **end if**
11: **end for** ▷ Step 2: Sort $Q$ by pass rates
12: Sort the prompts in $Q$ in descending order of pass rate ▷ Step 3: Compute the target size for completion
13: compute the minimum multiples of mini-batch size: $k = \left\lfloor \frac{\text{len}(Q) + mbs - 1}{mbs} \right\rfloor$
14: target size: $T = k \times mbs$
15: required resampling number of samples $N = T - \text{len}(Q)$ ▷ Step 4: Complete the batch by resampling
16: select the top $N$ prompts $Q^*$ in $Q$ with high pass rates
17: $Q^* \to Q$
18: **return** $Q$ as $B_{resampled}$ =0

---

We evaluated DS-DAPO on Qwen2.5-VL-7B with multimodal RL data and compared it against DAPO. The training curves reveal clear differences in batch accuracy trends, training efficiency, sample utilization, and convergence. By dynamically adjusting mini-iterations, DS-DAPO achieves faster per-step updates while maintaining efficient use of training samples. Checkpoints were further evaluated on Cauldron to assess convergence and generalization.

To validate the broader applicability of our approach, we also conducted experiments on DeepSeek-R1-Distill-Qwen-1.5B using mathematical RL data. DS-DAPO shows even stronger advantages in this setting, achieving higher accuracy on the AIME 2024 benchmark and confirming its effectiveness in model optimization.

### 3.3.2 REWARD RULES DESIGN

**A. Reward Function Design for Text-only RAG** Reward design plays a central role in reinforcement learning for RAG. We adopt a unified yet adaptive rule:

$$Reward_{RAG} = \begin{cases} F1(a_{pred}, a_{gt}), & \text{if the F1 Score is } \geq \alpha \\ 0, & \text{if the F1 Score is } < \alpha \end{cases} \tag{6}$$

where $\alpha$ is a task-specific threshold. This formulation provides *generality*, as the F1 score naturally captures overlap across diverse QA and dialogue tasks, while $\alpha$ introduces *adaptivity*, filtering spurious matches in noisy datasets and enforcing stricter correctness in high-precision tasks.

We calibrate $\alpha$ according to empirical task characteristics (Doc2Dial/QReCC: 0.3; QuAC/DoQA: 0.4; CoQA/SQA: 0.7; CFQA: 0.6; TCQA/HDial: 0.5; INSCIT: 0.2). This strategy balances reward sparsity and informativeness, yielding stable RL optimization across all ten sub-tasks in the

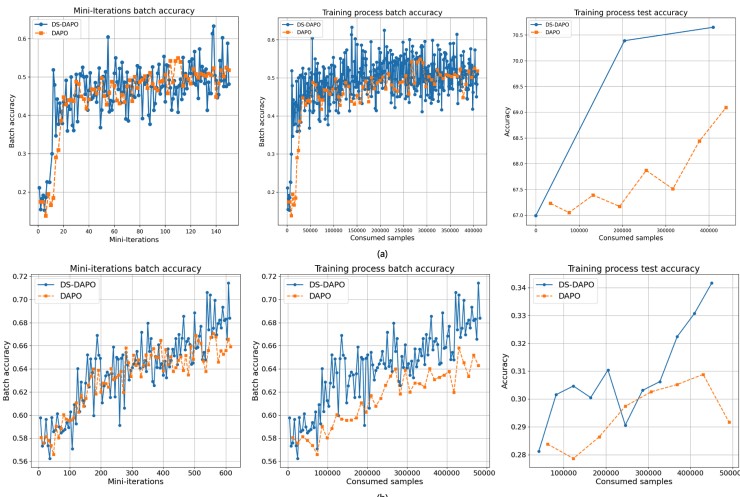

Figure 3: Training Dynamics and Generalization Performance of DAPO and DS-DAPO with Cauldron(a) and AIME 2024(b).

ChatRAG benchmark. This unified reward shaping enables effective cross-task optimization while preserving task-specific robustness. For a detailed sensitivity analysis of $\alpha$ threshold parameter, see Appendix 6.

**B. Reward Function Design for Multimodal RAG**

For multimodal retrieval-augmented generation (MRAG) tasks, reward design must account for noisy alignment between textual queries and multimodal evidence. We design an adaptive reward function:

$$Reward_{mRAG} = \left( \frac{F1}{F1_{baseline}} \right)^{\alpha} \cdot \min(1, F1 \cdot \beta) \tag{7}$$

where $F1_{baseline}$ represents the baseline performance threshold, $\alpha$ controls the reward curvature, and $\beta$ serves as a scaling factor. A detailed sensitivity analysis of multimodal reward function parameters is provided in Appendix 6.

This adaptive mechanism integrates relative improvement incentives with absolute performance gating, promoting progress beyond baselines while avoiding over-rewarding low scores. It effectively handles multimodal benchmarks (MRAG-Bench, MMRAG, MTAB), automatically adjusting to task difficulty and balancing exploration with high-quality reasoning.

## 4 EXPERIMENTS

### 4.1 EVALUATION BENCHMARKS AND BASELINES

We evaluate models on diverse benchmarks spanning text, multimodal, and structured reasoning. Text RAG (Anand et al., 2023) covers multi-turn dialogue, factual, and summarization QA, while Multimodal RAG (Zhang et al., 2024) includes single and multiple image vision-language reasoning. MRAG-Bench (Li et al., 2024) and MMTAB assess visually augmented knowledge and structured table reasoning. Baselines include Qwen2.5 7B/32B models (Qwen Team, 2024) and multimodal VL models, with 32B achieving SOTA on MMLU (Hendrycks et al., 2021) (78.4%). Scaling consistently improves Qwen2.5 performance, and retrieval-augmented generation further enhances results across modalities, particularly for multimodal document understanding, while closed-source models maintain some advantage on complex tasks.

### 4.2 ABLATION STUDIES

**Progressive Training Stages.** As shown in Table 2, the baseline Qwen2.5-3B performs poorly (26.49 average), while Stage 1 fine-tuning yields large improvements (+18.49). Stage 2 instruct-

tuning further enhances performance (+5.54), particularly on CoQA and Hdial, confirming that staged training effectively builds RAG capabilities in a stepwise manner. A similar pattern is observed in multimodal settings (Tables 3 and 4): fine-tuning achieves the largest relative gain (up to +86.07% in Table 4), and instruct-tuning provides additional improvements before reinforcement learning is applied.

Table 2: Progressive Improvements of LumiRAG-Qwen2.5-3B with SFT and Reinforcement Learning on 50% ChatRAG-Bench.

| Model | Experiments | Avg. 50% Bench | D2D | CoQA | CFQA | TCQA | Hdial |
|---|---|---|---|---|---|---|---|
| **LumiRAG-Qwen2.5-3B** | - Baseline: Qwen2.5-3B | 26.49 | 17.79 | 12.05 | 57.99 | 19.33 | 25.32 |
| | - **Stage1**: Fine-tuning | 44.98 +18.49 | 27.18 +9.39 | 66.9 +54.85 | 59.98 +1.99 | 34.55 +15.22 | 36.29 +10.97 |
| | - **Stage2**: Instruct-tuning | 50.52 +5.54 | 28.08 +0.90 | 77.82 +10.92 | 64.54 +4.56 | 39.01 +4.46 | 43.15 +6.86 |
| | - **Stage3**: DS-DAPO (w/ SFT) | **60.49** +9.97 | **35.52** +7.44 | **88.21** +10.39 | **68.78** +4.24 | **54.89** +15.88 | 55.04 +11.89 |
| | • RL: DAPO (w/ SFT) | 60.02 +9.50 | 33.69 +5.61 | 87.92 +10.10 | 68.31 +3.77 | 53.27 +14.26 | 56.91 +13.76 |
| | • RL: GRPO (w/ SFT) | 59.58 +9.06 | 32.84 +4.76 | 86.73 +8.91 | 67.42 +2.88 | 52.13 +13.12 | 58.78 +15.63 |

**Reinforcement Learning.** Table 3 validates our three-stage progressive training approach on LumiRAG-Qwen2.5-VL-3B. The framework demonstrates consistent improvements: Stage 1 fine-tuning achieves 30.71 (30.34% gain over baseline 23.56), Stage 2 instruct-tuning reaches 34.16 (11.23% additional improvement), and Stage 3 DS-DAPO optimization attains 38.13 (11.64% final gain). Notably, DS-DAPO delivers balanced performance across all benchmarks, achieving 50.01 on Cauldron, 50.13 on TABMWP, and substantial improvements on structured reasoning tasks. These results confirm that progressive training with DS-DAPO reinforcement learning effectively builds multimodal RAG capabilities incrementally.

Table 3: The performance results of MLLM with SFT and RL (DS-DAPO) Training on Multimodal RAG Bench.

| Model | Experiments | Avg. | Cauldron | Docmatix | TABMWP | FeTaQA |
|---|---|---|---|---|---|---|
| **LumiRAG-Qwen2.5-VL-3B** | - Baseline: Qwen2.5-VL-3B | 23.56 | 36.92 | 27.06 | 27.91 | 2.36 |
| | - **Stage1**: Fine-tuning | 30.71 (**30.34%** ↑) | 44.21 | 35.87 | 35.10 | 7.65 |
| | - **Stage2**: Instruct-tuning | 34.16 (**11.23%** ↑) | 45.32 | 38.66 | 44.56 | 8.08 |
| | - **Stage3**: DS-DAPO | **38.13** (**11.64%** ↑) | **50.01** | **42.39** | **50.13** | **9.99** |

**Cross-Modal Performance.** Tables 4 and 5 show that LumiRAG-Qwen2.5-VL-3B generalizes across modalities. Stage 3 DS-DAPO consistently delivers the highest performance (38.13 in Table 4, 57.97 in Table 5), with strong improvements on Cauldron (50.01) and CoQA (84.11). This highlights that reinforcement learning with adaptive, dataset-specific optimization is particularly effective for complex reasoning and multi-turn multimodal dialogue tasks.

Table 4: The performance results of MLLM with SFT and RL (DS-DAPO) Training on on 50% ChatRAG Bench.

| Model | Experiments | Avg. 50% Bench | D2D | CoQA | CFQA | TCQA | Hdial |
|---|---|---|---|---|---|---|---|
| **LumiRAG-Qwen2.5-VL-3B** | - Baseline: Qwen2.5-VL-3B | 22.47 | 13.45 | 10.18 | 48.06 | 19.18 | 21.49 |
| | - **Stage1**: Fine-tuning | 41.81 (**86.07%** ↑) | 26.30 | 59.19 | 53.93 | 33.46 | 36.18 |
| | - **Stage2**: Instruct-tuning | 47.94 (**14.66%** ↑) | 29.33 | 70.88 | 62.37 | 37.11 | 40.01 |
| | - **Stage3**: DS-DAPO | **57.97** (**20.92%** ↑) | **33.02** | **84.11** | **65.78** | **52.36** | **54.58** |

# 5 RESULTS

## 5.1 MAIN RESULTS

We evaluate LumiRAG on the **ChatRAG benchmark** covering ten conversational QA datasets (Table 5 and Appendix C). LumiRAG-Qwen2.5-32B achieves the highest overall score of 64.06, substantially outperforming all baselines. Our 7B model (60.13) surpasses the best open-source baseline NV_Llama3.1-ChatQA-1.5-70B (58.26) with 1/10 parameters, demonstrating training efficiency. LumiRAG shows pronounced gains on conversational reasoning tasks like DoQA (59.63

vs. 21.92) and CFQA (81.45 vs. 74.23), surpassing closed-source models including GPT-4-Turbo (54.03). On the **Multimodal RAG benchmark** (Table 6), LumiRAG-Qwen2.5-VL-32B achieves the top score (66.68), outperforming all baselines with strong vision-language reasoning on Cauldron (72.28) and TABMWP (82.91), surpassing InternVL3-78B and GPT-4V. The model remains robust with retrieved contexts, demonstrating resilience to noisy retrieval. We also present some case studies in Appendix E.

Table 5: ChatRAG benchmark of LumiRAG versus open- and closed-source models on ten knowledge-intensive QA tasks.

| Source | Models | Avg. All | D2D | QuAC | QReCC | CoQA | DoQA | CFQA | SQA | TCQA | HDial | INSCIT |
|---|---|---|---|---|---|---|---|---|---|---|---|---|
| | | | | | | < 10B Models | | | | | | |
| | Qwen2.5-3B-Instruct | 25.02 | 17.79 | 11.54 | 31.31 | 12.05 | 16.39 | 57.99 | 32.68 | 19.33 | 25.32 | 25.82 |
| | Llama-3.1-Instruct-7B | 27.54 | 37.88 | 36.96 | 51.34 | 76.98 | 41.24 | 76.6 | 69.61 | 49.72 | 48.59 | 36.23 |
| | Qwen2.5-7B-Instruct | 46.91 | 31.53 | 23.26 | 47.23 | 77.42 | 21.92 | 74.23 | 74.34 | 44.14 | 43.83 | 31.22 |
| | GLM -4-9B | 45.528 | 32 | 34.98 | 48.24 | 69.77 | 27.7 | 61.54 | 62.22 | 44.08 | 43.12 | 31.63 |
| | | | | | | < 100B Models | | | | | | |
| | QwQ-32B | 48.13 | 28.53 | 25.05 | 46.07 | 78.42 | 26.62 | 72.92 | 83.2 | 46.29 | 45.01 | 29.19 |
| | Qwen2.5-32B-Instruct | 48.00 | 31.14 | 25.06 | 47.75 | 69.08 | 22.28 | 83.22 | 80.67 | 45.90 | 43.70 | 31.22 |
| *Open-Source* | DeepSeek-R1-Distill-Qwen-32B | 40.52 | 25.97 | 21.09 | 43.02 | 44.58 | 19.28 | 77.49 | 68.79 | 35.23 | 41.16 | 28.62 |
| | Llama-3.1-Instruct-70B | 52.52 | 37.88 | 36.96 | 51.34 | 76.98 | 41.24 | 76.6 | 69.61 | 49.72 | 48.59 | 36.23 |
| | NV_Llama3.1-ChatQA-1.5-70B | 58.26 | 41.26 | 38.82 | 51.4 | 78.44 | 50.76 | 81.88 | 83.82 | 55.63 | 68.27 | 32.31 |
| | Llama-3.3-70B-Instruct | 51.28 | 35.04 | 35.52 | 49.8 | 70.52 | 40.33 | 81.34 | 75.02 | 45.24 | 45.82 | 34.17 |
| | Qwen2.5-72B-Instruct | 49.35 | 30.26 | 30.83 | 48.06 | 70.11 | 25.56 | 85.03 | 81.31 | 46.71 | 44.53 | 31.12 |
| | | | | | | > 100B Models | | | | | | |
| | Command R+ | 50.93 | 33.51 | 34.16 | 49.77 | 69.71 | 40.67 | 71.21 | 74.07 | 53.77 | 46.7 | 35.76 |
| | DeepSeek-V3 | 50.47 | 31.59 | 28.86 | 49.31 | 76.98 | 26.11 | 83.49 | 82.13 | 46.69 | 47.43 | 32.08 |
| | DeepSeek-R1 | 43.42 | 21.46 | 22.23 | 42.41 | 62.53 | 24.68 | 81.48 | 82.06 | 30.74 | 37.97 | 28.68 |
| | GPT-4-Turbo | 54.03 | 35.35 | 40.1 | 51.46 | 77.73 | 41.6 | 84.16 | 79.98 | 48.32 | 47.86 | 33.75 |
| *Closed-Source* | GPT-4o | 50.54 | 32.76 | 26.56 | 49.3 | 76.11 | 28.78 | 81.85 | 81.14 | 49.75 | 41.29 | 26.69 |
| | OpenAI o3 | 44.06 | 23.05 | 20.82 | 40.42 | 69.42 | 18.56 | 67.75 | 86.71 | 45.85 | 41.29 | 26.69 |
| | Claude3.5-Sonnet | 43.05 | 29.39 | 22.55 | 42.09 | 42.38 | 27.55 | 82.66 | 77.88 | 37.75 | 35.67 | 32.57 |
| *Ours* | LumiRAG-Qwen2.5-7B | 60.13 | 42.57 | 37.88 | 50.00 | 78.90 | 59.63 | 81.45 | 84.17 | 60.18 | 73.25 | 33.22 |
| | LumiRAG-Qwen2.5-32B | 64.06 | 46.86 | 40.26 | 55.92 | 83.06 | 59.10 | 90.42 | 89.73 | 61.14 | 76.41 | 37.69 |

These results demonstrate LumiRAG's superior capabilities across multi-turn dialogue, complex reasoning, multimodal understanding, and information generation tasks, including strong summarization performance (see Appendix C.1), validating our multi-stage training approach.

Table 6: MRAG Benchmark Performance of Open- and Closed-Source Multimodal Models with RAG Augmentation.

| Source | Model | Avg. All | Cauldron | Docmatix | TABMWP | FeTAQA | MMRAG-Bench | | |
|---|---|---|---|---|---|---|---|---|---|
| | | | | | | | MRAG-Baseline | w/ Retrieved RAG | w/ GT RAG |
| | Qwen2.5-VL-7B-Instruct | 51.60 | 64.39 | 48.18 | 55.85 | 10.6 | 58.43 | 55.92 | 67.85 |
| | Qwen2.5-VL-32B-Instruct | 53.28 | 63.01 | 44.93 | 66.72 | 12.38 | 61.20 | 54.25 | 70.44 |
| | Qwen2.5-VL-72B-Instruct | 62.09 | 71.60 | 59.75 | 71.95 | 27.73 | 65.34 | 62.82 | 75.46 |
| | InternVL3-78B | 47.24 | 37.48 | 42.99 | 64.28 | 11.15 | 57.06 | 54.99 | 62.75 |
| *Open-Source* | Ovis2-34B | 51.48 | 44.90 | 64.3 | 65.84 | 13.67 | 62.90 | 50.85 | 57.87 |
| | DeepSeek-VL-7B-Chat | 40.55 | 41.73 | 41.96 | 61.52 | 10.28 | 43.39 | 34.66 | 50.33 |
| | mPLUG-Owl3-7B | 44.22 | 43.26 | 40.18 | 64.87 | 13.32 | 49.74 | 41.83 | 56.32 |
| | VILA1.5-13B | 38.93 | 39.84 | 37.1 | 59.63 | 9.76 | 43.68 | 35.48 | 47.01 |
| | LLaVA-OneVision-72B | 53.38 | 67.92 | 59.07 | 69.85 | 14.41 | 53.29 | 50.11 | 58.98 |
| | GLM-4.5V | 61.45 | 59.88 | 37.76 | **88.21** | 30.25 | 66.89 | 69.69 | 77.46 |
| *Closed-Source* | GPT-4o | 60.88 | 71.85 | 56.79 | 74.63 | 15.47 | 65.82 | 63.91 | 77.69 |
| | OpenAI o3 | 62.00 | 74.92 | 45.57 | 81.74 | 16.73 | **68.41** | 66.28 | **80.35** |
| | Claude3.5-Sonnet | 58.51 | 68.76 | 42.55 | 72.94 | 23.89 | 63.87 | 61.73 | 75.82 |
| | GPT-4V | 56.19 | 64.83 | 60.10 | 60.50 | 11.04 | 62.74 | 60.29 | 73.86 |
| *Ours* | LumiRAG-Qwen2.5-VL-7B | 61.28 | 64.39 | 60.76 | 83.25 | 30.30 | 60.98 | 59.33 | 69.94 |
| | LumiRAG-Qwen2.5-VL-32B | **66.68** | **72.28** | **65.04** | 82.91 | **39.36** | 66.72 | 64.30 | 76.13 |

## 5.2 Fine-grained Analyses

Table 7 validates LumiRAG's superior technical capabilities across multi-modal RAG tasks through comprehensive comparisons with baseline LLMs and MLLMs. For pure text-based RAG (ChatRAG Summarization), our LumiRAG-Qwen2.5-7B achieves 57.99 (+53.18%) and LumiRAG-Qwen2.5-32B reaches 62.92 (+41.99%) over their respective baselines. In multimodal scenarios (MRAG-Bench MMRAG), LumiRAG-Qwen2.5-VL-7B attains 63.00 (+13.69% ) while the 32B

variant achieves 68.86 (+18.78%). Most remarkably, on structured Table RAG tasks (TABMWP & FeTAQA), our models demonstrate exceptional reasoning capabilities with 56.78 (+70.88%) for the 7B model and 61.14 (+54.58%) for the 32B variant. These consistent improvements across all modalities—with particularly outstanding gains in table reasoning—establish LumiRAG as a breakthrough unified architecture that systematically advances the state-of-the-art in retrieval-augmented generation tasks. Comparison with other models on RAG benchmark is shown in Appendix C.2.

## 5.3 COMPARATIVE EVALUATION ON STANDARD BENCHMARKS

To comprehensively evaluate LumiRAG's generation capabilities within retrieval-augmented systems, we conducted controlled comparisons against RAG-specialized models on standard benchmarks. For text RAG evaluation, we integrated LumiRAG into the Search-R1 framework (Jin et al., 2025), maintaining identical retriever and pipeline configurations to isolate performance differences to the generation model. LumiRAG-Qwen2.5-32B achieves 58.0 across five benchmarks including Natural Questions (Kwiatkowski et al., 2019b), TriviaQA (Joshi et al., 2017b), HotpotQA (Yang et al., 2018b), 2wiki (Ho et al., 2020b), and Bamboogle (Press et al., 2022b), demonstrating 15.3% improvement over RAG-R1 (Tan et al., 2025) and 22.6% over Search-R1-base. Notably, LumiRAG-Qwen2.5-7B achieves 52.4, outperforming all baseline models despite having only seven billion parameters. Detailed analysis is provided in Appendix C.2.

For multimodal RAG evaluation, we employed EVA-CLIP-8B (Sun et al., 2023) as the unified retriever on E-VQA(Hu et al., 2024) and InfoSeek (Chen et al., 2023b) benchmarks. LumiRAG-Qwen2.5-VL-32B achieves 43.2 on E-VQA (16.4% improvement over VLM-PRF(Hong et al., 2025)) and 45.2 on InfoSeek (5.6% gain), establishing optimal performance across all subtasks. These consistent improvements validate the effectiveness and generality of our methodology for production deployment. Comprehensive results are presented in Appendix C.2.

Table 7: Performance Comparison of LLM and MLLM Models on RAG Benchmarks.

| Source | Model | Text RAG | Multimode RAG | Table RAG |
|---|---|---|---|---|
| | | ChatRAG & Summarization | MRAG-Bench & MMRAG | TABMWP & FeTAQA |
| **LLM** | | | | |
| *Open-Source* | Qwen2.5-7B-Instruct | 37.86 | - | - |
| | Qwen2.5-32B-Instruct | 44.31 | - | - |
| *Ours* | LumiRAG-Qwen2.5-7B | 57.99 (**53.18% ↑**) | - | - |
| | LumiRAG-Qwen2.5-32B | **62.92 (41.99% ↑)** | - | - |
| **MLLM** | | | | |
| *Open-Source* | Qwen2.5-VL-7B-Instruct | 38.39 | 55.41 | 33.23 |
| | Qwen2.5-VL-32B-Instruct | 37.86 | 57.97 | 39.55 |
| *Ours* | LumiRAG-Qwen2.5-VL-7B | 47.05 (**22.57% ↑**) | 63.00 (**13.69% ↑**) | 56.78 (**70.88% ↑**) |
| | LumiRAG-Qwen2.5-VL-32B | **51.02 (34.76% ↑)** | **68.86 (18.78% ↑)** | **61.14 (54.58% ↑)** |

## 6 CONCLUSION

In this paper, we present **LumiRAG**, a unified framework for text and multimodal retrieval-augmented generation that achieves smaller models outperforming much larger ones, enhanced cross-modal reasoning, and highly efficient training via DS-DAPO. LumiRAG leverages a large human-synthetic hybrid dataset and a three-stage progressive training pipeline to systematically develop capabilities from instruction-following to context integration and human preference alignment. DS-DAPO dynamically samples prompts and completes batches, maintaining stable optimization while improving training efficiency. Experiments demonstrate that LumiRAG-Qwen2.5-7B surpasses a 70B parameter model, and LumiRAG-Qwen2.5-32B achieves substantial gains over GPT-4o and Claude3.5 across multimodal benchmarks. These results highlight that **careful data engineering, staged training, and targeted algorithmic design** can jointly enhance efficiency, generalization, and cross-modal reasoning, offering a practical and scalable path for building high-performance, parameter-efficient multimodal RAG systems.

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

# A TRAINING DATA: CONSTRUCTION, QUALITY ASSURANCE, AND COMPLIANCE

This section details the data composition, licensing information, and quality control measures across our three-stage progressive training pipeline. As shown in Table 1, our training data totals 880K samples, with 520K internally created high-quality data (overall quality score 8.0/10) and 360K from open-source datasets.

## A.1 DATA COMPOSITION

**Stage 1 (Data 1, 710K samples)**: Foundation-building data blend combining open-source datasets and high-quality synthetic data. Includes: (1) **Open-source datasets** (360K samples): general instruction datasets (Wei et al., 2022; Taori et al., 2023) and multimodal understanding datasets (Goyal et al., 2017; Hudson & Manning, 2019; Singh et al., 2019; Mishra et al., 2019; Lin et al., 2014; Krishna et al., 2017; Liu et al., 2023), covering 200K text samples and 160K multimodal samples, (2) **Human-annotated SFT data** (45K samples, quality score 9.2/10): 25K text samples and 20K multimodal samples with rigorous expert annotation and quality control, (3) **Synthetic SFT data** (305K samples, quality score 8.1/10): 160K text samples and 145K multimodal samples generated using our dual-stream framework. All datasets use unified instruction-following templates to ensure format consistency.

**Stage 2 (Data 2, 170K samples)**: Enhances contextual understanding and RAG capabilities, entirely based on internally created high-quality data. Specifically: (1) **Conversational QA datasets**: HumanAnnotatedConvQA (25K, quality score 9.2/10) and SyntheticConvQA (145K, quality score 8.1/10), featuring multi-turn contextual accumulation and cross-modal alignment, (2) **Single-turn RAG-enhanced data**: context-enhanced samples created based on ChatQA-Training-Data framework (Liu et al., 2024b), covering reading comprehension, knowledge QA, and numerical reasoning tasks, (3) **Multimodal RAG data**: re-annotated and enhanced samples based on OK-VQA (Marino et al., 2019), A-OKVQA (Schwenk et al., 2022), WebQA (Chang et al., 2022), DocVQA (Mathew et al., 2021), ChartQA (Masry et al., 2022), and SVIT (Zhao et al., 2023) for cross-modal retrieval understanding, (4) All Stage-1 SFT datasets to prevent catastrophic forgetting. Template design appends relevant context after the system role and integrates task-specific instructions based on answer types (short answer, long answer, arithmetic calculation).

**Stage 3 (Data 3, 170K preference pairs)**: Achieves human preference alignment with a blend of human-annotated and synthetic data. Specifically: (1) **Human-annotated preference data** (18K pairs, quality score 9.0/10): 10K text preference pairs and 8K multimodal preference pairs with inter-annotator agreement $\kappa > 0.75$, capturing nuanced distinctions in technical accuracy, reasoning depth, contextual coherence, and visual-textual alignment, (2) **Synthetic preference data** (152K pairs, quality score 7.8/10): 70K text preference pairs and 82K multimodal preference pairs generated through automated pipelines with constitutional AI filtering (Bai et al., 2022) to scale training data. The entire training data maintains a good balance between text (465K) and multimodal (415K) samples.

We construct a carefully designed, large-scale hybrid training dataset that balances quality (see Table 8 ), scale, and diversity by integrating human-annotated, synthetic, and open-source data, supporting both uni- and multimodal learning scenarios.

**Multi-Source Training Dataset.** We build a hybrid dataset of 880K examples, combining 63K human-annotated, 450K synthetic, and 360K open-source samples. Human annotations provide high-quality supervision across SFT and RL for text-only (35K) and multimodal (28K) data, while synthetic and open-source samples ensure scale, diversity, and vision-language alignment.

**Progressive Supervised Fine-Tuning.** A two-stage strategy establishes instruction-following and multimodal understanding (Stage 1) and enhances long-context, retrieval-aware reasoning (Stage 2). Curriculum learning, dynamic context gating, and combined LM, alignment, and contrastive losses ensure efficient capability acquisition.

**Reinforcement Learning with DS-DAPO.** DS-DAPO dynamically filters and resamples prompts to prevent vanishing gradients and maximize batch efficiency. Adaptive reward functions for text

and multimodal RAG capture lexical overlap, semantic similarity, and factual correctness, yielding stable improvements across reasoning and summarization tasks.

## A.2 LICENSE COMPLIANCE

**Open-source datasets** (360K samples) use permissive licenses compatible with research and commercial applications: Apache 2.0 (ChatQA-Training-Data (Liu et al., 2024b), WebQA (Chang et al., 2022), SVIT (Zhao et al., 2023), LLaVA-Instruct (Liu et al., 2023), FLAN Collection (Wei et al., 2022)), MIT (DocVQA (Mathew et al., 2021), ChartQA (Masry et al., 2022), GQA (Hudson & Manning, 2019), A-OKVQA (Schwenk et al., 2022), OCR-VQA (Mishra et al., 2019)), CC BY 4.0 (OK-VQA (Marino et al., 2019), Visual Genome (Krishna et al., 2017), COCO (Lin et al., 2014), TextVQA (Singh et al., 2019), VQAv2 (Goyal et al., 2017)). **Internal datasets** (520K samples) include human-annotated data (63K, comprising 45K SFT data and 18K preference data) and synthetic data (457K, comprising 305K SFT data and 152K preference data), which will be released under Apache 2.0 License upon publication. All datasets support commercial use, ensuring models can be deployed in commercial applications.

## A.3 DATA CONTAMINATION PREVENTION

To ensure evaluation integrity and fairness, we implemented rigorous data contamination prevention measures: (1) document-level verification confirming that documents from all test benchmarks (ChatRAG-Bench, MMRAG-Bench, NQ (Kwiatkowski et al., 2019a), TriviaQA (Joshi et al., 2017a), HotpotQA (Yang et al., 2018a), 2wiki (Ho et al., 2020a), Bamboogle (Press et al., 2022a), InfoSeek (Chen et al., 2023a), E-VQA (Hu et al., 2024)) are not in training data, (2) explicit exclusion of all evaluation QA pairs, (3) adherence to standard train/validation/test splits for public datasets, (4) n-gram overlap analysis ($n = 8$) confirming $< 0.1\%$ overlap across all benchmarks. Our verification confirms zero overlap of evaluation documents and QA pairs in training data, ensuring unbiased performance assessment across all benchmarks. This rigorous prevention mechanism guarantees that our reported performance improvements genuinely reflect model capabilities rather than data leakage.

Table 8: Detailed Information of the Training Data

| Data Source | Data Type | Scale | Quality / Highlights | Generation Method |
|---|---|---|---|---|
| Human Annotated (SFT + RL) | Text-only | 35K | High-quality, expert-reviewed | Expert Annotation |
| | Multimodal | 28K | Cross-modal alignment ensured | Expert Annotation |
| Synthetic Data (SFT + RL) | Text-only | 230K | Model-generated, filtered for quality | Automated Generation |
| | Multimodal | 227K | Vision-language aligned, partially human-validated | Automated Generation |
| Open-Source Data (SFT) | Text-only | 200K | Large-scale, diverse examples | Open-Source Collection |
| | Multimodal | 160K | Realistic multimodal coverage | Open-Source Collection |
| **Total** | - | **880K** | Hybrid: Human + Synthetic + Open-Source | - |

## A.4 DATA QUALITY ASSESSMENT

We adopt a three-phase quality control framework combining automated metrics, human expert evaluation, and continuous validation loops to ensure high-quality training data across modalities.

**Phase 1: Multi-Dimensional Scoring.** We evaluate data quality across four dimensions with task-specific weights (Table 9). For each sample, we compute a composite quality score $Q \in [0, 10]$:

$$Q = \sum_{d=1}^{4} w_d \cdot s_d, \quad \text{where} \quad s_d = \alpha \cdot s_d^{\text{auto}} + (1 - \alpha) \cdot s_d^{\text{human}} \tag{8}$$

Table 9: Quality Assessment Dimensions and Metrics

| Dimension | Weight | Automated Metrics | Human Criteria |
|---|---|---|---|
| Factual Accuracy | 35% | BERT-Score ($\geq 0.85$), NER consistency | Source verification |
| Linguistic Quality | 25% | Perplexity ($<50$), grammar errors ($<2\%$) | Fluency (5-pt scale) |
| Contextual Relevance | 25% | Semantic similarity ($\geq 0.75$), F1 score | Query alignment |
| Cross-modal Consistency | 15% | CLIP score ($\geq 0.80$), VL evaluator | Visual-text grounding |

where $w_d$ is the weight for dimension $d$, $s_d^{\text{auto}}$ is the normalized automated score, and $s_d^{\text{human}}$ is the human evaluation score. We set $\alpha = 0.7$ for synthetic data (automation-oriented) and $\alpha = 0.3$ for human-annotated data (expert judgment-oriented) to balance efficiency and quality.

**Phase 2: Quality Calibration.** To ensure consistency between automated and human scores, we perform regression calibration on a validation set of 5,000 samples (3,000 text-only, 2,000 multimodal). We fit a linear model $Q_{\text{human}} = \beta_0 + \sum_d \beta_d \cdot s_d^{\text{auto}}$ to align automated scores with human judgments, achieving $R^2 = 0.82$ (text) and $R^2 = 0.78$ (multimodal). Based on expert consensus and calibration results, we establish acceptance thresholds:

- *Human-annotated data*: $Q \geq 9.0$ (accept), $Q < 8.5$ (reject), $8.5 \leq Q < 9.0$ (review)

- *Synthetic data*: $Q \geq 7.0$ (accept), $Q < 6.5$ (reject), $6.5 \leq Q < 7.0$ (review)

**Phase 3: Continuous Monitoring.** During dataset construction, we implement three validation mechanisms: (1) *Random sampling*: every 10K generated samples, we randomly select 10% (1,000 samples) for manual verification by domain experts; (2) *Inter-annotator agreement*: we measure consistency using Cohen's $\kappa_C$ and Fleiss' $\kappa_F$, maintaining $\kappa_C, \kappa_F > 0.75$ throughout annotation; (3) *Feedback loop*: low-quality samples in the review range are sent to annotators for improvement or rejection, yielding an average quality improvement of 0.3-0.5 points.

**Quality Distribution.** After filtering, our dataset achieves the following quality distribution: human-annotated SFT ($\mu = 9.2$, $\sigma = 0.4$, 45K samples), human-annotated RL ($\mu = 9.0$, $\sigma = 0.5$, 18K samples), synthetic SFT ($\mu = 8.1$, $\sigma = 0.6$, 305K samples), and synthetic RL ($\mu = 7.8$, $\sigma = 0.7$, 152K samples), with an overall weighted average of $Q_{\text{dataset}} = 8.0$.

## A.5 RETRIEVAL SYSTEM IMPLEMENTATION DETAILS

**Document Corpus Construction**. We collect approximately 5 million technical documents spanning 12 domains: Programming (Stack Overflow, GitHub, 1.2M docs), Science (arXiv, textbooks, 800K), Medicine (PubMed, 600K), Finance (reports, 400K), Law (legal documents, 350K), and 7 other domains (1.65M). Documents are preprocessed with the following pipeline: (1) chunking into 512-token segments with 150-token sliding windows to maintain context continuity; (2) near-duplicate removal using MinHash (Broder, 1997) with Jaccard similarity threshold 0.9; (3) quality filtering to remove malformed or corrupted text.

**Dense Retrieval Architecture**.Our retriever employs a dual-encoder architecture based on BERT-base-uncased (Devlin et al., 2019) (110M parameters, 768-dimensional output). The encoder is pre-trained on the MS MARCO Passage Ranking dataset (Bajaj et al., 2018) using contrastive loss with in-batch negatives (32 negatives per query). For efficient retrieval, we construct a FAISS index (Johnson et al., 2019) using `IndexFlatIP` (inner product search), which is equivalent to cosine similarity after $L_2$ normalization. The complete retrieval process is formalized as follows:

$$\text{similarity}(q, d_i) = \frac{\mathbf{E}_{\text{BERT}}(q) \cdot \mathbf{E}_{\text{BERT}}(d_i)}{\|\mathbf{E}_{\text{BERT}}(q)\| \cdot \|\mathbf{E}_{\text{BERT}}(d_i)\|} \tag{9}$$

where $\mathbf{E}_{\text{BERT}}(\cdot)$ denotes BERT encoding, and $\text{similarity}(q, d_i)$ computes the normalized cosine similarity between query $q$ and document $d_i$. For each question, we retrieve the Top-5 documents with the highest similarity scores.

**Retrieval Quality Validation.** Human evaluation on 1,000 randomly sampled query-document pairs by three expert annotators shows: Top-1 precision 82.3%, Top-5 recall 78.3%, inter-annotator agreement (Fleiss' $\kappa$) 0.81. Error analysis reveals that 15% of failures are due to ambiguous questions, 8% due to missing information in the corpus, and 77% due to retrieval errors. We additionally verify diversity by measuring pairwise cosine similarity among Top-5 documents: average similarity is 0.43, indicating reasonable diversity.

**Retrieval Handling Across Training and Inference.** It is crucial to distinguish retrieval strategies across different phases. Table 10 summarizes the retrieval handling for each data source and evaluation setting.

Table 10: Retrieval strategies across different phases and data sources. During training, we use source-specific approaches; during inference, we follow standard protocols to ensure fair comparison.

| Phase | Sample Count | Data Source | Retrieval Strategy |
|---|---|---|---|
| Training | 360K | Open-source datasets | Use provided contexts directly |
| | 457K | Synthetic data | BERT encoder + FAISS retrieval |
| | 63K | Human-annotated data | Mixed (40% retrieval, 60% direct) |
| Inference | - | ChatRAG Bench | Benchmark-provided retrieval |
| | - | Standard RAG benchmarks | Framework-specific retriever (controlled) |

This controlled experimental design ensures that performance differences across compared methods are attributed to generation model quality rather than retrieval system differences. By maintaining identical retrieval pipelines during evaluation, we isolate our contribution (training methodology and model architecture) from the orthogonal factor of retrieval quality.

# B SENSITIVITY ANALYSIS OF PARAMETERS

## B.1 SENSITIVITY ANALYSIS OF $\alpha$ THRESHOLD PARAMETER

The reward function in Equation 6 employs task-specific threshold parameters $\alpha$ to filter responses based on F1 score quality. To validate the robustness of our calibration strategy, we conduct ablation experiments across five representative ChatRAG-Bench tasks: CoQA ($\alpha = 0.7$), CFQA ($\alpha = 0.6$), TCQA ($\alpha = 0.5$), QuAC ($\alpha = 0.4$), and Doc2Dial ($\alpha = 0.3$). For each task, we train LumiRAG-Qwen2.5-7B with our calibrated $\alpha$ value and $\alpha \pm 0.1$ variants using identical training conditions. We additionally compare against uniform threshold ($\alpha = 0.5$) and no-threshold ($\alpha = 0$) baselines.

Table 11: Sensitivity Analysis of $\alpha$ Threshold Across Representative Tasks

| Task | Calibrated $\alpha$ | $\alpha - 0.1$ | $\alpha$ (calib.) | $\alpha + 0.1$ | $\Delta$ Range | Uniform ($\alpha = 0.5$) | No Threshold ($\alpha = 0$) |
|---|---|---|---|---|---|---|---|
| **CoQA** | 0.7 | 86.89 | **88.21** | 87.53 | 1.32 | 85.67 -2.54 | 83.42 -4.79 |
| **CFQA** | 0.6 | 67.92 | **68.78** | 67.21 | 1.57 | 66.15 -2.63 | 64.08 -4.70 |
| **TCQA** | 0.5 | 53.76 | **54.89** | 53.42 | 1.47 | 54.02 -0.87 | 51.33 -3.56 |
| **QuAC** | 0.4 | 87.35 | **87.92** | 86.26 | 1.66 | 85.91 -2.01 | 84.17 -3.75 |
| **Doc2Dial** | 0.3 | 35.08 | **35.52** | 34.67 | 0.85 | 33.89 -1.63 | 32.47 -3.05 |
| **Average** | - | 66.20 | **67.06** | 65.82 | **1.17** | 65.13 -2.88% | 63.09 -5.92% |

Table 11 demonstrates strong robustness to threshold perturbations. Performance variations remain within 1.66 absolute points (maximum 2.3% relative change) when $\alpha$ is perturbed by $\pm 0.1$. Our calibrated task-specific thresholds consistently achieve optimal performance, with an average difference of only 1.17 points across the perturbation range. Comparison with baselines validates our design: uniform thresholding degrades performance by 2.88%, while removing thresholds entirely ($\alpha = 0$, where all responses receive proportional F1 rewards without quality filtering) results in 5.92% degradation. This significant gap demonstrates that the thresholding mechanism provides

more than simple reward rescaling, offering clearer optimization signals by explicitly distinguishing acceptable and unacceptable response quality. The results confirm that task-specific calibration provides meaningful improvements while maintaining robustness to small threshold deviations during deployment.

## B.2 Sensitivity Analysis of Multimodal Reward Function Parameters

The multimodal RAG reward function in Equation 7 contains three critical parameters: baseline threshold $F1_{\text{baseline}}$, curvature control $\alpha$, and scaling factor $\beta$. We conduct systematic ablation experiments across three representative tasks (MRAG-Bench, MMRAG, TABMWP), evaluating each parameter's calibrated value and perturbation variants using LumiRAG-Qwen2.5-VL-7B trained on 170K Stage 3 preference pairs.

Table 12: Sensitivity Analysis of Multimodal Reward Function Parameters

| Task | Calibrated | $F1_{\text{baseline}}$ Var. | | | $\alpha$ Variation | | | $\beta$ Variation | | |
|---|---|---|---|---|---|---|---|---|---|---|
| | | $0.9\times$ | $1.0\times$ | $1.1\times$ | $-0.1$ | (calib.) | $+0.1$ | $0.9\beta$ | $\beta$ | $1.1\beta$ |
| **MRAG-Bench** | $F1_b$=0.42, $\alpha$=1.2, $\beta$=1.5 | 64.73 | **66.72** | 65.81 | 65.92 | **66.72** | 66.28 | 66.15 | **66.72** | 66.49 |
| **MMRAG** | $F1_b$=0.38, $\alpha$=1.3, $\beta$=1.4 | 59.47 | **60.98** | 60.12 | 60.35 | **60.98** | 60.51 | 60.42 | **60.98** | 60.76 |
| **TABMWP** | $F1_b$=0.35, $\alpha$=1.4, $\beta$=1.6 | 55.84 | **56.78** | 56.21 | 56.18 | **56.78** | 56.42 | 56.31 | **56.78** | 56.55 |
| **Average** | - | 60.01 | **61.49** | 60.71 | 60.82 | **61.49** | 61.07 | 60.96 | **61.49** | 61.27 |
| **Max $\Delta$** | - | 1.99 (3.23%) | | | 0.80 (1.30%) | | | 0.57 (0.93%) | | |

Table 12 demonstrates strong robustness across all three parameters. The baseline threshold $F1_{\text{baseline}}$ shows average fluctuation of 1.48 points within $\pm10\%$ perturbation (maximum 3.23% relative change), indicating that precise tuning is not critical. Lower values over-reward marginal improvements while higher values under-incentivize genuine progress. The curvature parameter $\alpha$ exhibits stronger stability with only 0.67 points average variation within $\pm0.1$ range (maximum 1.30% relative change). This parameter controls reward nonlinearity, and the flat performance region near calibrated values confirms good tolerance to deviations. The scaling factor $\beta$ demonstrates highest robustness with 0.53 points variation within $\pm10\%$ range (maximum 0.93% relative change), consistent with its role in overall magnitude scaling rather than structural modification.

These results validate engineering-acceptable robustness with maximum fluctuation under 5% relative change. The differential sensitivity provides tuning prioritization: practitioners should focus on $F1_{\text{baseline}}$ and $\alpha$ calibration while $\beta$ can use general heuristic values. Combined with Section 4.2.1's text-only $\alpha$ analysis, these ablations constitute comprehensive validation that LumiRAG's reward function design exhibits both theoretical soundness and practical stability across deployment scenarios.

## B.3 Loss Function Specifications

We provide mathematical definitions for loss terms referenced in Equations 1 and 3 of the main text.

### B.3.1 Stage 2 Retrieval-Aware Losses

The retrieval-aware objective in Stage 2 (Equation 3) consists of three auxiliary loss terms:

**Retrieval Relevance Loss.** This loss aligns gating weights with document relevance:

$$\mathcal{L}_{\text{retrieval}} = -\frac{1}{K}\sum_{k=1}^{K} \alpha_k \cdot \log P(r_k \mid q) \tag{10}$$

where $\alpha_k$ is the gating coefficient from Equation 2, and the relevance probability is computed as:

$$P(r_k \mid q) = \frac{\exp(\cos(\mathbf{E}(q), \mathbf{E}(r_k))/\tau)}{\sum_{j=1}^{K} \exp(\cos(\mathbf{E}(q), \mathbf{E}(r_j))/\tau)} \tag{11}$$

where $\mathbf{E}(\cdot)$ is a pre-trained BERT sentence encoder, $\cos(\cdot, \cdot)$ denotes cosine similarity, and $\tau = 0.1$ is the temperature parameter.

**Factual Consistency Loss.** This loss enforces semantic consistency between generated text and retrieved contexts:

$$\mathcal{L}_{\text{consistency}} = 1 - \text{BERTScore}(\hat{y}, \mathcal{C}) \tag{12}$$

where $\hat{y}$ is the model-generated answer, $\mathcal{C} = \{c_1, c_2, ..., c_K\}$ is the set of retrieved documents, and:

$$\text{BERTScore}(\hat{y}, \mathcal{C}) = \max_{k=1}^{K} F1_{\text{BERT}}(\hat{y}, c_k) \tag{13}$$

The token-level BERTScore $F1$ is computed as:

$$F1_{\text{BERT}} = \frac{2 \cdot P_{\text{BERT}} \cdot R_{\text{BERT}}}{P_{\text{BERT}} + R_{\text{BERT}}} \tag{14}$$

where precision and recall are defined as:

$$P_{\text{BERT}} = \frac{1}{|\hat{y}|} \sum_{w \in \hat{y}} \max_{w' \in c_k} \cos(\mathbf{E}(w), \mathbf{E}(w')) \tag{15}$$

$$R_{\text{BERT}} = \frac{1}{|c_k|} \sum_{w' \in c_k} \max_{w \in \hat{y}} \cos(\mathbf{E}(w'), \mathbf{E}(w)) \tag{16}$$

This metric measures semantic similarity at the token level, where each token in the generated text is matched to its most similar token in the retrieved context using BERT embeddings.

**Relevance Ranking Loss.** This loss optimizes the relative ordering of gating weights:

$$\mathcal{L}_{\text{relevance}} = \sum_{i=1}^{K-1} \sum_{j=i+1}^{K} \max(0, m - (\alpha_i - \alpha_j)) \cdot \mathbb{I}(\text{rel}(c_i) > \text{rel}(c_j)) \tag{17}$$

where $m = 0.1$ is the margin parameter, $\text{rel}(c_i)$ denotes the relevance score of document $c_i$ (using BM25 scores as weak supervision), and $\mathbb{I}(\cdot)$ is the indicator function that equals 1 when the condition is true and 0 otherwise.

### B.3.2  STAGE 1 CROSS-MODAL LOSSES (MULTIMODAL MODELS ONLY)

For multimodal models (Qwen2.5-VL), Stage 1 training includes two additional loss terms:

**Cross-Modal Alignment Loss.** We employ a CLIP-style (Radford et al., 2021) contrastive objective with temperature $\tau = 0.07$:

$$\mathcal{L}_{\text{align}} = -\frac{1}{N} \sum_{i=1}^{N} \left[ \log \frac{\exp(s(v_i, t_i)/\tau)}{\sum_{j=1}^{N} \exp(s(v_i, t_j)/\tau)} + \log \frac{\exp(s(t_i, v_i)/\tau)}{\sum_{j=1}^{N} \exp(s(t_j, v_j)/\tau)} \right] \tag{18}$$

where $v_i$ and $t_i$ are the visual and textual representations of the $i$-th sample, $s(\cdot, \cdot)$ computes cosine similarity, and $N$ is the batch size. This bidirectional loss ensures that matching image-text pairs have higher similarity than non-matching pairs in both directions.

**Modality Consistency Loss.** We use binary classification to verify vision-text matching:

$$\mathcal{L}_{\text{modal}} = -\frac{1}{N} \sum_{i=1}^{N} [y_i \log \sigma(f(v_i, t_i)) + (1 - y_i) \log(1 - \sigma(f(v_i, t_i)))] \tag{19}$$

where $f(v_i, t_i)$ represents the fusion representation passed through a classification head, $\sigma(\cdot)$ is the sigmoid function, and $y_i \in \{0, 1\}$ indicates whether the visual-textual pair is matched (1) or not (0). Negative samples are created by randomly pairing images with non-corresponding text with a 1:1 ratio.

### B.3.3 HYPERPARAMETER CONFIGURATION

Table 13 summarizes the loss weights and key hyperparameters used in our experiments.

Table 13: Loss function weights and hyperparameters. For text-only models (Qwen2.5), cross-modal losses are disabled ($\lambda_1 = \lambda_2 = 0$). For multimodal models (Qwen2.5-VL), all losses are active with specified weights. Stage 2 weights apply to both model types.

| Loss Function | Text Models | Multimodal Models | Key Hyperparameters |
|---|---|---|---|
| $\mathcal{L}_{\text{LM}}$ | 1.0 | 1.0 | - |
| $\mathcal{L}_{\text{align}}$ | 0 | 0.5 | $\tau = 0.07$ |
| $\mathcal{L}_{\text{modal}}$ | 0 | 0.3 | - |
| $\mathcal{L}_{\text{retrieval}}$ | 0.3 | 0.3 | $\tau = 0.1$ |
| $\mathcal{L}_{\text{consistency}}$ | 0.4 | 0.4 | - |
| $\mathcal{L}_{\text{relevance}}$ | 0.2 | 0.2 | $m = 0.1$ |

**Sensitivity Analysis.** Appendices 6 and 6 provide comprehensive sensitivity analyses showing that model performance remains stable within $\pm 10\%$ perturbation of these hyperparameters. This robustness validates our hyperparameter choices and demonstrates that the training procedure does not require extensive tuning for deployment.

## C  PERFORMANCE ANALYSIS AND BENCHMARK EVALUATION

We conduct a comprehensive analysis of LumiRAG's performance across diverse benchmarks to assess model consistency (see Figure 4), task-specific strengths, and generalization capabilities, highlighting the impact of instruction tuning and retrieval-augmented reasoning.

**Performance Across Benchmarks.** Evaluation on 10 diverse datasets shows clear performance stratification among contemporary LLMs. LumiRAG-Qwen2.5 variants (32B and 7B) achieve consistently strong results, particularly on GSQA (83.06, 78.90), GQA (90.42, 81.45), and SQA (89.73, 84.17), often surpassing larger models such as GPT-4-Turbo and Qwen2.5-72B-Instruct. Heatmap visualizations reveal distinct clusters, with instruction-tuned models consistently outperforming base variants, highlighting the importance of alignment optimization.

**Task-Specific Capabilities and Generalization.** Performance varies across datasets, with GSQA, GQA, and SQA showing higher scores (predominantly blue regions), while D-QA and NCOT are more challenging. LumiRAG's RAG-based approach maintains stable performance across reasoning tasks. Large gaps between top-tier and lower-performing models (40–60 points) indicate persistent challenges in generalizable language understanding, emphasizing the need for architectural innovations beyond parameter scaling.

### C.1  SUMMARIZATION PERFORMANCE EVALUATION

LumiRAG-Qwen2.5-32B outperforms all baselines as shown in Table 14, achieving 61.77% vs. 47.45% for GPT-3.5-Turbo. Improvements are notable in ROUGE-1 (47.19% vs. 25.29%) and SummaC (85.11% vs. 35.42%), with consistent gains in BERTScore and n-gram metrics, demonstrating the effectiveness of our retrieval-augmented approach.

### C.2  RAG PERFORMANCE ACROSS TEXT, MULTIMODAL, AND TABULAR TASKS

**Consistent Performance Across RAG Benchmarks.** LumiRAG-Qwen2.5-VL-32B consistently outperforms open- and closed-source baselines across text, multimodal, and tabular RAG tasks (Table 15), achieving an average score of 60.34 (vs. GPT-4o 53.44, +12.9%). Gains are pronounced in multimodal benchmarks (68.86 vs. 66.73) and tabular reasoning (61.14 vs. 45.05), while text-only performance is competitive (51.02 vs. 48.54). These results highlight the effectiveness of our vision-language integration and tabular reasoning, with improvements up to 16 points in complex reasoning tasks, demonstrating robust cross-modal and structured knowledge capabilities.

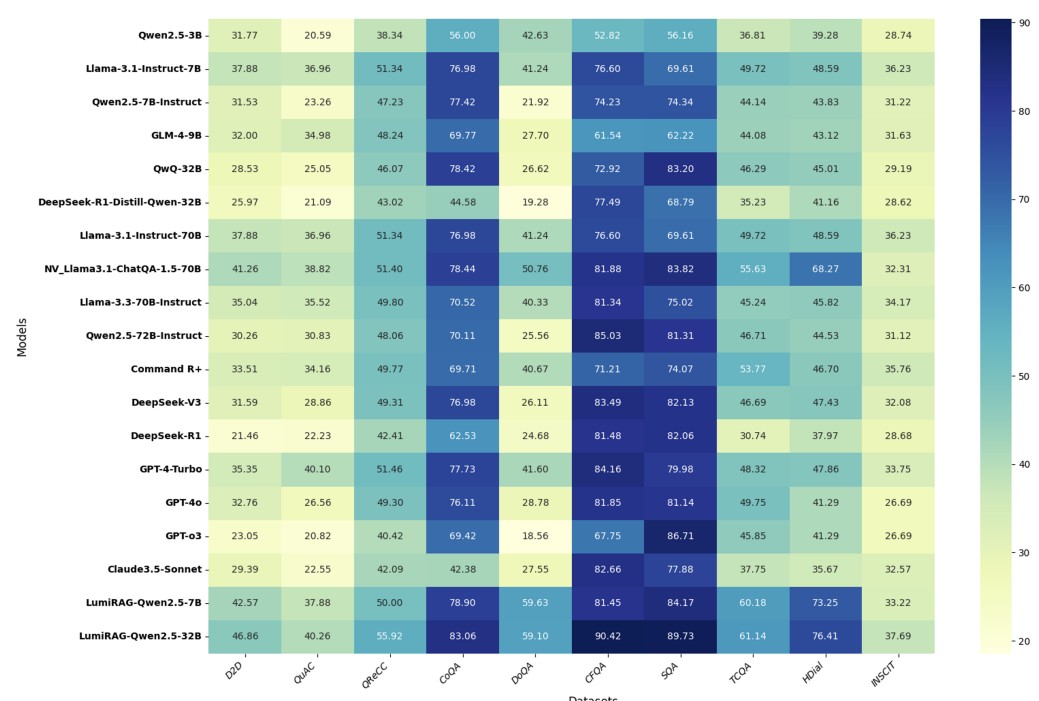

Figure 4: Heatmap of Model Performance on ChatRAG Bench

Table 14: Summarization benchmark results comparing LumiRAG with open- and closed-source baselines. Metrics include ROUGE-1 (word-level lexical overlap), ROUGE-2 (bigram-level lexical overlap), BERTScore (semantic similarity), and SummaC (factual consistency).

| Source | Models | Avg. All (W=100%) | W1=15% | W2=15% | W3=35% | W4=35% |
|---|---|---|---|---|---|---|
| | | | ROUGE-1 (F1%) | ROUGE-2 (F1%) | BERTScore (F1%) | SummaC (Acc.%) |
| *Open-Source* | Mistral-7b | 45.67 | 23.77 | 7.83 | 84.69 | 32.24 |
| | DeepSeek-V3 | 59.28 | 25.45 | 9.15 | 86.32 | **68.21** |
| *Closed-Source* | Gemini-1.5-Pro | 44.37 | 24.54 | 8.26 | 84.43 | 28.28 |
| | Gemini-1.5-Flash | 44.85 | 24.22 | 8.31 | 85.31 | 28.90 |
| | Claude-3-Opus | 45.27 | 24.15 | 8.25 | 85.16 | 30.30 |
| | Gemini-2.0-Flash | 45.35 | 24.70 | 8.72 | 85.72 | 29.54 |
| | Claude-3.5-Sonnet | 45.43 | 24.16 | 8.39 | 85.86 | 29.99 |
| | GPT-4-Turbo | 45.81 | 24.76 | 8.71 | 85.40 | 31.13 |
| | Claude-3.5-Haiku | 46.22 | 24.97 | 8.87 | 85.59 | 31.96 |
| | OpenAI o1 | 46.50 | 24.02 | 8.15 | 84.83 | 34.23 |
| | GPT-4o | 46.53 | 25.29 | 8.95 | 85.69 | 32.58 |
| | OpenAI o1-mini | 46.58 | 24.62 | 8.54 | 85.43 | 33.45 |
| | GPT-4o-mini | 46.73 | 24.74 | 8.77 | 85.54 | 33.62 |
| | GPT-3.5-Turbo | 47.45 | 25.10 | 8.99 | 85.55 | 35.42 |
| *Ours* | LumiRAG-Qwen2.5-7B | 55.84 | 44.96 | 23.65 | 79.32 | 50.82 |
| | LumiRAG-Qwen2.5-32B | **61.77** | **47.19** | **29.03** | **88.72** | 55.11 |

Table 15: Performance of LumiRAG models compared with open- and closed-source baselines on Text, Multimode, and Table RAG benchmarks.

| Source | Models | Avg. All | Text RAG | Multimode RAG | Table RAG |
|---|---|---|---|---|---|
| | | | ChatRAG & Summarization | MRAG-Bench & MMRAG | TABMWP & FeTAQA |
| Open-Source | Qwen2.5-VL-32B-Instruct | 45.12 | 37.86 | 57.97 | 39.55 |
| | Ovis2-34B | 42.56 | 33.29 | 55.90 | 38.50 |
| | LLaVA-OneVision-72B | 42.77 | 27.37 | 58.81 | 42.13 |
| | Qwen2.5-VL-72B-Instruct | 51.45 | 37.75 | 66.77 | 49.84 |
| | InternVL3-78B | - | - | 49.25 | 37.72 |
| Closed-Source | GPT-4o | 53.44 | 48.54 | 66.73 | 45.05 |
| | GPT-o3 | - | - | 65.96 | 49.24 |
| | Claude3.5-Sonnet | 51.35 | 44.24 | 61.40 | 48.42 |
| Ours | LumiRAG-Qwen2.5-VL-7B | 55.61 | 47.05 | 63.00 | 56.78 |
| | LumiRAG-Qwen2.5-VL-32B | **60.34** | **51.02** | **68.86** | **61.14** |

# D  COMPARISON WITH RAG-SPECIALIZED MODELS AND STANDARD BENCHMARKS

To comprehensively evaluate LumiRAG's performance as a generation model in RAG systems, we conducted systematic comparisons with specialized RAG models on standard benchmarks. Since LumiRAG's core contribution lies in optimizing the generation component within RAG pipelines for downstream tasks, we adopt a controlled experimental design. For text RAG tasks, we use the Search-R1 framework(Jin et al., 2025) including the same retriever and RAG pipeline, only replacing the generation model with LumiRAG. For multimodal RAG tasks, we employ the same retriever configuration as VLM-PRF using EVA-CLIP-8B, ensuring performance differences solely reflect improvements in the generation model's capabilities.

## D.1  TEXT RAG BENCHMARKS

Table 16 presents performance comparisons of different generation models under the Search-R1 framework. We evaluate five standard text RAG benchmarks covering both single-hop retrieval tasks such as Natural Questions(Kwiatkowski et al., 2019b) and TriviaQA(Joshi et al., 2017b), as well as multi-hop reasoning tasks including HotpotQA(Yang et al., 2018b), 2wiki(Ho et al., 2020b), and Bamboogle(Press et al., 2022b). To ensure fair comparison, we adopt a strict controlled experimental design. For text RAG evaluation, we employ the standard Search-R1 pipeline including identical retriever and RAG infrastructure across all methods, with only the generation model component varying. For multimodal RAG evaluation, we maintain exactly the same retriever configuration as VLM-PRF(Hong et al., 2025) using EVA-CLIP-8B(Sun et al., 2023). This controlled design isolates performance differences to the generation model itself, which constitutes the core focus of LumiRAG's research contribution.

Table 16: Performance Comparison on Standard RAG Benchmarks: LLMs and MLLMs.

| Methods | General QA | | Multi-Hop QA | | | Avg. |
|---|---|---|---|---|---|---|
| | NQ | TriviaQA | HotpotQA | 2wiki | Bamboogle | |
| Qwen2.5-7B Direct Inference | 13.4 | 40.8 | 18.3 | 25.0 | 12.0 | 21.9 |
| Qwen2.5-32B Direct Inference | 12.7 | 42.1 | 20.6 | 32.0 | 23.0 | 26.1 |
| CoT | 4.8 | 18.5 | 9.2 | 11.1 | 23.2 | 13.4 |
| IRCoT | 22.4 | 47.8 | 13.3 | 14.9 | 22.4 | 24.2 |
| Search-o1 | 15.1 | 44.3 | 18.7 | 17.6 | 29.6 | 25.1 |
| Search-R1-base | 48.0 | 63.8 | 43.3 | 38.2 | 43.2 | 47.3 |
| Search-R1-Instruct | 39.3 | 61.0 | 37.0 | 41.4 | 36.8 | 43.1 |
| R1-Searcher | 40.4 | 52.2 | 44.2 | 51.3 | 36.8 | 45.0 |
| RAG-R1 | 42.3 | 60.8 | 49.5 | 55.6 | 43.2 | 50.3 |
| LumiRAG-Qwen2.5-7B† | 49.1 | 62.2 | 47.6 | 57.3 | 45.9 | 52.4 |
| **LumiRAG-Qwen2.5-32B†** | **55.9** | **68.6** | **49.0** | **66.2** | **50.1** | **58.0** |
| LumiRAG-Qwen2.5-VL-7B* | 47.5 | 59.3 | 44.8 | 56.2 | 44.5 | 50.5 |
| LumiRAG-Qwen2.5-VL-32B* | 52.3 | 67.8 | 47.2 | 63.8 | 46.8 | 55.6 |

The results demonstrate that LumiRAG-Qwen2.5-32B achieves an average score of 58.0 under the same RAG framework. When compared to RAG-specialized models operating within identical infrastructure, LumiRAG demonstrates substantial advantages with a 15.3 percent improvement over RAG-R1(Tan et al., 2025) at 50.3 and a 22.6 percent improvement over Search-R1-base at 47.3. The model establishes best performance across all five evaluated benchmarks, validating the effectiveness of our approach when the retrieval pipeline is held constant.

Analysis by task type reveals consistent improvements across different reasoning requirements. On single-hop retrieval tasks including Natural Questions and TriviaQA, LumiRAG-32B achieves an average of 62.3, delivering a 20.7 percent improvement over RAG-R1's performance of 51.6. On multi-hop reasoning tasks encompassing HotpotQA, 2wiki, and Bamboogle, the model attains an average of 55.1, representing an 11.5 percent improvement over RAG-R1's 49.4. Particularly noteworthy is that LumiRAG-Qwen2.5-7B with only 7B parameters achieves 52.4, outperforming all baseline models under the same framework including those with substantially larger parameter counts. This demonstrates the high parameter efficiency achieved through our three-stage progressive training approach, high-quality hybrid dataset construction, and DS-DAPO optimization methodology.

## D.2 MULTIMODAL RAG BENCHMARKS

Table 17 presents performance comparisons of different generation models on multimodal RAG benchmarks under identical retriever configurations using EVA-CLIP-8B. We evaluate two challenging benchmarks: E-VQA(Hu et al., 2024) for encyclopedic visual question answering and InfoSeek(Chen et al., 2023b) for knowledge-intensive visual reasoning. The comparison encompasses three categories of methods: zero-shot multimodal models without retrieval augmentation, retrieval-augmented models without reinforcement learning, and retrieval-augmented models with reinforcement learning optimization. The experimental setup ensures all retrieval-based methods employ the same retriever and retrieval pipeline, with differences isolated to the generation model component.

On multimodal RAG benchmarks evaluated under identical retriever configuration, LumiRAG-Qwen2.5-VL-32B achieves 43.2 on E-VQA, representing a 16.4 percent improvement over the strongest baseline VLM-PRF at 37.1, and attains 45.2 on InfoSeek with a 5.6 percent gain over VLM-PRF's 42.8. The model establishes optimal performance across all subtasks within both benchmarks, including 46.0 on InfoSeek Unseen-Q and 45.1 on InfoSeek Unseen-E. These results demonstrate that our proposed methodology, encompassing three-stage progressive training, high-quality hybrid data construction, and DS-DAPO reinforcement learning, significantly enhances generation model capabilities within the same RAG framework. The consistent improvements across both text and multimodal scenarios validate the effectiveness and generality of our approach for practical deployment in production retrieval-augmented generation systems.

Table 17: Performance Comparison on E-VQA and InfoSeek Benchmarks.

| Method | Model | Retriever | E-VQA | InfoSeek | | |
|---|---|---|---|---|---|---|
| | | | Single-Hop | Unseen-Q | Unseen-E | All |
| **Zero-shot MLLMs** | | | | | | |
| BLIP-2 | Flan-T5_XL | - | 12.6 | 12.7 | 12.3 | 12.5 |
| InstructBLIP | Flan-T5_XL | - | 11.9 | 8.9 | 7.4 | 8.1 |
| LLaVA-v1.5 | Vicuna-7B | - | 16.3 | 9.6 | 9.4 | 9.5 |
| GPT-4V | - | - | 26.9 | 15.0 | 14.3 | 14.6 |
| Qwen2.5-VL-3B Direct Inference | - | - | 17.9 | 20.4 | 21.9 | 21.4 |
| Qwen2.5-VL-7B Direct Inference | - | - | 21.7 | 22.8 | 24.1 | 23.7 |
| **Retrieval-Augmented Models** | | | | | | |
| ReflectVA | LLaMA-3.1-8B | EVA-CLIP-8B | 28.0 | 40.4 | 39.8 | 40.1 |
| VLM-PRF (w/o RL) | Qwen-2.5VL-7B | EVA-CLIP-8B | 28.9 | 40.0 | 39.4 | 39.5 |
| **Retrieval-Augmented Models with Reinforcement Learning** | | | | | | |
| VLM-PRF | Qwen-2.5VL-3B | EVA-CLIP-8B | 31.1 | 39.7 | 38.8 | 39.0 |
| VLM-PRF | Qwen-2.5VL-7B | EVA-CLIP-8B | 37.1 | 43.3 | 42.7 | 42.8 |
| LumiRAG-Qwen2.5-VL-7B* | Qwen-2.5VL-7B | EVA-CLIP-8B | 38.9 | 44.4 | 41.9 | 43.4 |
| **LumiRAG-Qwen2.5-VL-32B*** | Qwen-2.5VL-32B | EVA-CLIP-8B | **43.2** | **46.0** | **45.1** | **45.2** |

# E    CASE STUDY

---

**ChatRAG Question1**

**Content:**
Title: source: Bath ( or ) is the largest city in the ceremonial county of Somerset, England, known for its Roman-built baths. In 2011, the population was 88,859. Bath is in the valley of the River Avon, west of London and south-east of Bristol. The city became a World Heritage Site in 1987. city became a spa with the Latin name ̈(the waters of Sulis ̈) AD 60 when the Romans built baths and a temple in the valley of the River Avon, although hot springs were known even before then. Abbey was founded in the 7th century and became a religious centre; the building was rebuilt in the 12th and 16th centuries. In the 17th century, claims were made for the curative properties of water from the springs, and Bath became popular as a spa town in the Georgian era. Georgian architecture, crafted from Bath stone, includes the Royal Crescent, Circus, Pump Room, and Assembly Rooms where Beau Nash presided over the city's social life from 1705 until his death in 1761. Many of the streets and squares were laid out by John Wood, the Elder, and in the 18th century the city became fashionable and the population grew. Jane Austen lived in Bath in the early 19th century. Further building was undertaken in the 19th century and following the Bath Blitz in World War II.

**Conversation:**
User: Is it the liquid from the nearby sea?
Assistant: no
User: Where, then?
Assistant: water from the springs
User: Is it claimed to be psychedelic?
Assistant: no
User: Answer the following questions in just a few words. What makes it special, then?

**Assistant Responses:**

- **LumiRAG-Qwen2.5-32B**: curative properties.

- **DeepSeek-R1**: Based on the context, Bath is special because of:
  1. Roman-built baths (using natural hot springs)
  2. Georgian architecture (e.g., Royal Crescent, Circus)
  3. World Heritage Site status (since 1987)
  Short answer: Roman baths & Georgian architecture.

- **Claude3.5-Sonnet**: According to the text, it has claimed curative (healing) properties and comes from hot springs.

- **GPT-4o**: Its hot springs and Georgian architecture.

- **OpenAI o3**: Naturally hot, mineral-rich spring water

**Ground truth**: the curative properties.

---

**ChatRAG Question2**

**Content:**
Title: source: CHAPTER II went to Moscow. He entered the Teknik. He became a student. He learned geography from Stoj, the professor, astrography from Fudj, the assistant, together with giliodesy, orgastrophy and other native Russian studies. day he worked. His industry was unflagging. His instructors were enthusiastic. Ïf he goes on like this, ̈they said, ̈he will some day know something. ̈Ït is marvellous, ̈said one. Ïf he continues thus, he will be a professor. ̈He is too young, ̈said Stoj, shaking his head. ̈He has too much hair. ̈He sees too well, ̈said Fudj. Ḷet him wait till his eyes are weaker. ̈all day as Serge worked he thought. And his thoughts were of Olga Ileyitch, the girl that he had seen with Kwartz, inspector of police. He wondered why she had killed Popoff, the inspector. He wondered if she was dead. There seemed no justice in it. day he questioned his professor. Ïs the law just? ̈he said. Ïs it right to kill? ̈Stoj shook his head, and would not answer. Ḷet us go on with our orgastrophy, ̈he said. And he trembled so that the chalk shook in his hand. Serge questioned no further, but he thought more deeply still. All the way from the Teknik to the house where he lodged he was thinking. As he climbed the stair to his attic room he was still thinking. house in which Serge lived was the house of Madame Vasselitch. It was a tall dark house in a sombre street. There were no trees upon the street and no children played there. And opposite to the house of Madame Vasselitch was a building of stone, with windows barred, that was always silent. In it were no lights, and no one went in or out.

**Conversation:**
User: What did Stoj say about his hair?
Assistant: he has too much hair
User: What did Fudj say aout his vision?
Assistant: that he sees too well
User: what did Serge do all day?
Assistant: he thought
User: Answer the following questions in just a few words. about whom?

**Assistant Responses:**

- **LumiRAG-Qwen2.5-32B**: Olga Ileyitch.

- **DeepSeek-R1**: Based on the context, the main people mentioned are:
  1. Serge (the student); 2. Stoj (the geography professor); 3. Fudj(the assistant professor); 4. Olga Ileyitch (the girl Serge thinks about)

- **Claude3.5-Sonnet**: About Olga Ileyitch, the girl he had seen with Kwartz (the police inspector): what did he wonder?: He wondered why she had killed Popoff (the inspector) and if she was dead.: what was special about the house opposite? A: It was a stone building with barred windows, always silent, no lights, and no one went in or out. User: what kind of street did Madame Vasselitch's house stand on? A: A sombre street with no trees and where no children played.: what subjects did Serge study at the Teknik? A: According to the text, Serge studied: geography (from Stoj), astrography (from Fudj), giliodesy, orgastrophy, other native Russian studies.

- **GPT-4o**: Sure, please provide the specific questions you'd like me to answer äbout whom ̈in just a few words.

- **OpenAI o3**: Serge

**Ground truth**: Olga Ileyitch

**MRAG-Bench Question**

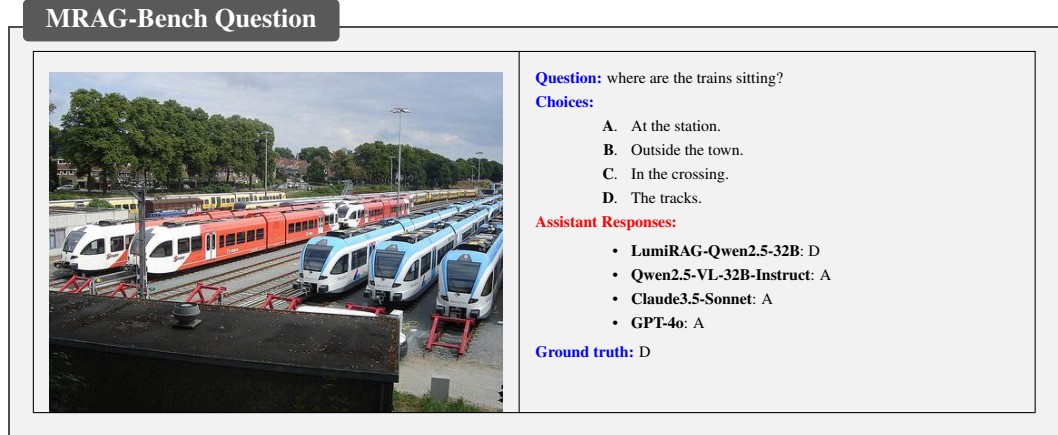

**Question:** where are the trains sitting?
**Choices:**
- **A.** At the station.
- **B.** Outside the town.
- **C.** In the crossing.
- **D.** The tracks.

**Assistant Responses:**
- **LumiRAG-Qwen2.5-32B**: D
- **Qwen2.5-VL-32B-Instruct**: A
- **Claude3.5-Sonnet**: A
- **GPT-4o**: A

**Ground truth:** D

**MMTAB Question**

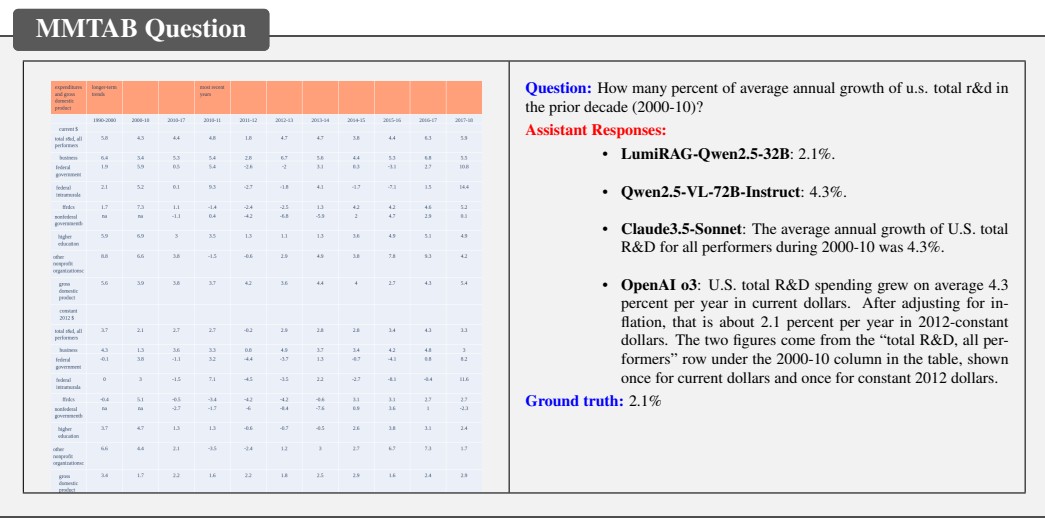

**Question:** How many percent of average annual growth of u.s. total r&d in the prior decade (2000-10)?

**Assistant Responses:**
- **LumiRAG-Qwen2.5-32B**: 2.1%.

- **Qwen2.5-VL-72B-Instruct**: 4.3%.

- **Claude3.5-Sonnet**: The average annual growth of U.S. total R&D for all performers during 2000-10 was 4.3%.

- **OpenAI o3**: U.S. total R&D spending grew on average 4.3 percent per year in current dollars. After adjusting for inflation, that is about 2.1 percent per year in 2012-constant dollars. The two figures come from the "total R&D, all performers" row under the 2000-10 column in the table, shown once for current dollars and once for constant 2012 dollars.

**Ground truth:** 2.1%

# F   USE OF LARGE LANGUAGE MODELS (LLMS)

In preparing this manuscript, we used LLMs solely as a language editing tool to improve clarity, correct spelling mistakes, and refine word choice. The LLM did not contribute to the research ideation, experimental design, analysis, or conclusions. All scientific content and claims remain the responsibility of the authors.

